# Deep Learning for Fully Automatic Tumor Segmentation on Serially Acquired Dynamic Contrast-Enhanced MRI Images of Triple-Negative Breast Cancer

**DOI:** 10.3390/cancers15194829

**Published:** 2023-10-02

**Authors:** Zhan Xu, David E. Rauch, Rania M. Mohamed, Sanaz Pashapoor, Zijian Zhou, Bikash Panthi, Jong Bum Son, Ken-Pin Hwang, Benjamin C. Musall, Beatriz E. Adrada, Rosalind P. Candelaria, Jessica W. T. Leung, Huong T. C. Le-Petross, Deanna L. Lane, Frances Perez, Jason White, Alyson Clayborn, Brandy Reed, Huiqin Chen, Jia Sun, Peng Wei, Alastair Thompson, Anil Korkut, Lei Huo, Kelly K. Hunt, Jennifer K. Litton, Vicente Valero, Debu Tripathy, Wei Yang, Clinton Yam, Jingfei Ma

**Affiliations:** 1Department of Imaging Physics, The University of Texas MD Anderson Cancer Center, Houston, TX 77030, USA; zxu5@mdanderson.org (Z.X.);; 2Department of Breast Imaging, The University of Texas MD Anderson Cancer Center, Houston, TX 77030, USA; 3Department of Breast Medical Oncology, The University of Texas MD Anderson Cancer Center, Houston, TX 77030, USA; 4Department of Clinical Research Imaging, The University of Texas MD Anderson Cancer Center, Houston, TX 77030, USA; 5Department of Biostatistics, The University of Texas MD Anderson Cancer Center, Houston, TX 77030, USA; 6Section of Breast Surgery, Baylor College of Medicine, Houston, TX 77030, USA; 7Department of Bioinformatics & Computational Biology, The University of Texas MD Anderson Cancer Center, Houston, TX 77030, USA; 8Department of Pathology, The University of Texas MD Anderson Cancer Center, Houston, TX 77030, USA; 9Department of Breast Surgical Oncology, The University of Texas MD Anderson Cancer Center, Houston, TX 77030, USA

**Keywords:** deep learning, tumor segmentation, triple-negative breast cancer

## Abstract

**Simple Summary:**

Quantitative image analysis of cancers requires accurate tumor segmentation that is often performed manually. In this study, we developed a deep learning model with a self-configurable nnU-Net for fully automated tumor segmentation on serially acquired dynamic contrast-enhanced MRI images of triple-negative breast cancer. In an independent testing dataset, our nnU-Net-based deep learning model performed automated tumor segmentation with a Dice similarity coefficient of 93% and a sensitivity of 96%.

**Abstract:**

Accurate tumor segmentation is required for quantitative image analyses, which are increasingly used for evaluation of tumors. We developed a fully automated and high-performance segmentation model of triple-negative breast cancer using a self-configurable deep learning framework and a large set of dynamic contrast-enhanced MRI images acquired serially over the patients’ treatment course. Among all models, the top-performing one that was trained with the images across different time points of a treatment course yielded a Dice similarity coefficient of 93% and a sensitivity of 96% on baseline images. The top-performing model also produced accurate tumor size measurements, which is valuable for practical clinical applications.

## 1. Introduction

Triple-negative breast cancer (TNBC) is an aggressive subtype of breast cancer, representing approximately 15% of all breast cancers and contributing to approximately 40% of breast cancer-related deaths [1]. Neoadjuvant systemic therapy followed by surgery is the standard-of-care treatment for TNBC. However, the responses of patients to neoadjuvant systemic therapy vary, and only approximately 50% of patients achieve a pathological complete response, which is a useful surrogate marker for favorable long-term clinical outcomes. Given the aggressive nature of TNBC and substantial variability in pathologic complete response rates, noninvasive imaging methods for accurate tumor characterization and early prediction of tumor response to therapy will be highly valuable. Quantitative image analyses of tumors are increasingly used for early detection of cancer [2], accurate tumor localization and staging [3], and treatment response assessment [4], or prediction [5].

An important step in quantitative image analyses is tumor segmentation. The most commonly used method of tumor segmentation is manual contouring and annotation by experienced radiologists. However, this process is labor-intensive and tedious, as well as being susceptible to human errors and inter-reader variations [6]. To overcome these challenges, computer-aided diagnosis algorithms have been developed. Such algorithms are usually model-based and involve active contouring [7,8,9], automated thresholding [10,11], region growing [12,13], or combinations of methods [14]. However, these methods are typically applicable only under specific assumptions since their optimization depends on pre-defined constraint thresholds. 

Deep learning techniques, especially convolutional neural networks [3], have been explored for their potential in delineating images or multidimensional features, including automated tumor segmentation [15,16]. A deep learning model [17] trained with over 45,000 mammograms outperformed a previous computer-aided diagnosis system that relied on designed features and selected seed points. U-Net, a dedicated convolutional neural network for medical imaging segmentation, has gained popularity [18]. An implementation using a two-stage U-Net demonstrated accurate breast tumor segmentation across multiple datasets [19]. According to a recent review [20], three out of six referenced studies of breast cancer segmentation were based on U-Net. 

Dynamic contrast-enhanced (DCE) MRI is capable of measuring contrast agent kinetics and has shown high sensitivity in detecting breast cancer [21]. The kinetic texture, which represents contrast enhancement characteristics, can be employed to segment regions of tissue with similar vascular properties [22,23]. Early methods of using DCE-MRI for tumor segmentation can be categorized as the atlas-based methods [24,25], cluster methods [26], and classifier-based methods [27]. Recent studies have shown that deep learning-based methods can offer superior tumor segmentation performance, potentially by integrating both kinetic characteristics of the signals and tumor texture information (e.g., voxel/pixel based, including tumor shape and background tissue homogeneity). Using fully convolutional networks, a hierarchical convolutional neural network framework was developed to perform segmentation [28]. Another deep learning approach is to compose a multiple-components U-Net framework to transform the segmentation into a multi-classification task, such as to separate tumors, fibroglandular tissue [29], and other tissues. DeepMedic [30] is one widely-accepted framework that efficiently computes the model parameters via a configurable multi-resolution pathway. It was originally developed for automated brain lesion segmentation, but has since been adapted for breast tumor studies [31,32]. Other deep learning frameworks, such as SegNet [31,33] and ResNet [3,34,35], also achieved good results in automated segmentation.

However, the success rate of the translation of these models trained from one dataset to a different and independent dataset is still very limited. The technical challenge is likely caused by the methods’ configuration and parameters optimization that are needed for the dataset diversity. Adapting a model iteratively requires extensive knowledge and effort, which increases the complexity of the clinical workflow. Another challenge is the lack of high-quality datasets for the model training of a specific disease population. MRI datasets are typically much larger in image size but smaller in population size than sonogram or mammogram datasets. The availability of DCE-MRI data specific to TNBC patients is even more limited and poses another barrier to train a functional model. 

In this work, we aimed to develop a model for automated segmentation of TNBC on DCE-MRI images. The deep learning framework we employed to build our model is nnU-Net [36], which is based on the standard U-Net structure but offers a unique feature of automated hyperparameter optimization. In nnU-Net, some model parameters are empirically derived using ten datasets from the Medical Segmentation Decatholon [37] while the remaining parameters are customized with the application-specific training datasets. We hypothesize that the nnU-Net framework, with a self-configuring segmentation model that has demonstrated broad success over a variety of datasets and image modalities, could provide accurate automated segmentation of TNBC using DCE-MRI images. To overcome the challenge of the limited datasets, we combined the data acquired serially throughout a patient’s treatment course and hypothesized that tumor progression over different time points will improve the segmentation performance compared to using the data from a single time point. Specifically, we trained and tested models over multiple semiquantitative maps from DCE-MRI images. The models were systematically evaluated in terms of the Dice similarity coefficient (DSC) and sensitivity. Our findings show that the subtraction between pre-contrast and post-contrast images provided the best performance. Using the model trained from data across all time points over the treatment course, we achieved a median DSC of 93% and sensitivity of 96% over our independent testing dataset across all time points.

## 2. Materials and Methods

This study was approved by the Institutional Review Board (IRB) of The University of Texas MD Anderson Cancer Center and was part of an ongoing IRB-approved prospective clinical trial (NCT02276443) of patients with stage I-III TNBC who were being monitored for responses to neoadjuvant systemic therapy. This study followed the ethical guidelines set out in the Declaration of Helsinki, and written informed consent was obtained from each participant.

### 2.1. Dataset

A total of 301 patients with biopsy-confirmed stage I-III TNBC were included in this study. The data inclusion criteria were identical to and described in a previously published work [38]. The imaging protocol for each patient included DCE-MRI, which was acquired at multiple time points during a patient’s treatment course: at baseline (BL), after two cycles (C2), and after four cycles (C4) of neoadjuvant systemic therapy. Among all the patients, 299 had BL scans, 221 had C2 scans, and 272 had C4 scans. Data from patients with inflammatory breast cancer, failed manual tumor segmentation due to technical issues, or a complete response to treatment without any visible residual enhancing lesions were excluded from model training. In total, 744 datasets (285 from BL, 207 from C2, and 252 from C4) were used for this study. Patients’ demographic and clinical characteristics for these datasets are presented in Table 1.

### 2.2. Image Acquisition

The DCE-MRI images were acquired using a 3T GE 750w MR scanner (GE Healthcare, Waukesha, WI, USA) and an eight-channel bilateral phased array breast coil. The imaging protocol utilized a three-dimensional (3D) T1-weighted DISCO [39] sequence with intravenous bolus injection of contrast agent (Gadovist, Bayer HealthCare, Whippany, NJ, USA) at a rate of 2 mL/s and a dose of 0.1 mL/kg, followed with a saline flush. The imaging parameters included an acquisition matrix size of 320 × 320, a field-of-view of 300 × 300 mm, a slice thickness of 3.2 mm, a slice gap of −1.6 mm, a TR of 6 ms, a TE of 1.1/2.3 ms, a flip angle of 12°, and a temporal resolution of approximately 12 s. The number of slices ranged from 112 to 192, and the number of temporal phases ranged from 32 to 64.

### 2.3. Data Curation

DCE-MRI images and binary masks were first zero-padded along both sides of the imaging volume to 192 slices, and then the full-field-of-view images were fed for model training without any cropping. Two breast radiologists with 6 years of experience (R.M.M.) and 11 years of experience (S.P.) manually segmented the tumors in consensus using an in-house MATLAB-based software (The MathWorks, Natick, MA, USA). These manually segmented tumors are shown by the reference masks in Figures 2 and 3. The manual segmentation was performed on a subtraction image obtained by subtracting the pre-injection phase (the initial time frame) from the early phase (the frame at approximately 2.5 min after injection). Voxels of high contrast uptake between phases were identified as tumors, and voxels of signal void from biopsy clips or tumor necrosis were excluded. In cases where multiple tumors were present, only the dominant one was labeled at BL, and the same tumor was followed at C2 and C4 if images were acquired at those time points. 

In addition, we calculated and used the following three semiquantitative maps: positive enhancement integral (PEI) [40], signal enhancement ratio (SER) [41], and maximum slope of increase (MSI). These maps [42] were calculated on an AW Server using the software provided by the vendor (v3.2, GE Healthcare, Milwaukee, WI, USA). 

To evaluate the impact of central necrosis and biopsy clips on segmentation accuracy, we compared models with their inclusion and exclusion. Unless noted otherwise, the default segmentation used for the constructed models described below in Section 2.4 relates to models with all necrosis and clips excluded (Mk_Excl). The tumor masks including necrosis and clips (Mk_Incl) were generated by filling the central void voxels within Mk_Excl automatically. Among all 285 BL cases, 38 cases were excluded as their Mk_Incl failed to include all voxels of necrosis and clips, resulting in 247 cases for model development. 

### 2.4. Automatic Segmentation Framework

We used the default original configuration of nnU-Net (Appendix A, nnU-Net model training configuration and procedure) without any major architectural changes [36]. During the model training, the input data were first preprocessed to extract the data fingerprint, which included median shapes, signal intensity distribution, spacing distribution, and modality. The preprocessing steps included cropping the images to non-zero regions to improve the computational efficiency. Then, the rule-based parameters, including batch size, resample strategy, intensity normalization, and network topology, were derived through a dictionary lookup approach. The nnU-Net framework also used fixed parameters that were pre-trained from different applications and hardcoded for all new studies, including the learning rate, optimizer, number of epochs, choice of activation function, and loss function. Finally, empirical parameters were calculated on the basis of the ensemble of 2D and 3D results and the integration of the models from five-fold cross-validation, which determined the details in inference and post-processing.

A total of 10 nnU-Net models (Table 2) were constructed using different combinations of input images. The first model utilized only subtraction image data from the BL dataset and was named nnU-Net_BL. The second, third, and fourth models were created using PEI, SER, and MSI data from the BL dataset and were named nnU-Net_PEI, nnU-Net_SER, and nnU-Net_MSI, respectively. The fifth model, nnU-Net_Comb, was generated using concatenating subtraction image, PEI, SER, and MSI data from the BL dataset to form an additional data dimension. These five models were designed to identify the most sensitive and accurate imaging metric from DCE-MRI images acquired at the same time point. The sixth and seventh models were trained using subtraction image data from the C2 and C4 datasets and were named nnU-Net_C2 and nnU-Net_C4, respectively. The eighth model nnU-Net was created by combining cohorts at three time points (BL, C2, and C4) and was named nnU-Net_3tpt. The nnU-Net_BL, nnU-Net_C2, nnU-Net_C4, and nnU-Net_3tpt models were evaluated to determine the accuracy of the models at different time points. The ninth and tenth models were with exclusion and inclusion of central necrosis and biopsy clips and were named nnU-Net_Excl and nnU-Net_Incl, respectively. 

Each dataset was randomly divided into development and testing sets at a 5:1 ratio. Each development set was further split for five-fold cross-validation at a ratio of 4:1 for training and validation. The development and testing sets of nnU-Net_3tpt were composed separately by merging corresponding sets from nnU-Net_BL, nnU-Net_C2, and nnU-Net_C4.

Upon completion of training, the models were ensembled by averaging the softmax probabilities from each fold of cross-validation. The resulting ensembled model was used for inference on independent testing data. The training was performed in both 2D and 3D models. During inference, the ensemble of both 2D and 3D prediction was generated by performing voxel-wise majority vote. 

All training was performed using an NVIDIA DGX1 system with dual 20-core Intel Xeon E5-2698 2.2-GHz CPUs, 512 GB of DDR4 RAM, and eight NVIDIA Tesla V100 32-GB GPUs with a total of 256 GB of GPU memory (NVIDIA, Santa Clara, CA, USA). The software environment included an Ubuntu Linux 18.4.6 operating system, Python 3.8.12, CUDA 11.1, cuDNN 7.6.5, and TensorFlow 2.8.0.

### 2.5. Statistical Analysis

The performance of each model was evaluated using four overlap-based segmentation metrics, true positive, false negative, false positive, and true negative, using the manually labeled mask as the reference standard on a per-subject basis. The true positive, false negative, false positive, and true negative are defined as the percentage of the number of voxels within the union between the reference and predicted masks, of the number of voxels within the reference mask but outside the predicted mask, the number of voxels within the predicted mask but outside the reference mask, and the number of background voxels outside the reference mask, respectively. The DSC and sensitivity were calculated and averaged across all subjects as the metrics for overall performance [43].

Given the non-normal distribution of the results, the within-group results of subject-based DSC and sensitivity were summarized using interquartile and median calculations; the paired two-sample comparison was performed with Wilcoxon signed-rank test and the unpaired two-sample comparison was performed with Wilcoxon rank-sum test; the multiple comparison was performed using Kruskal–Wallis test. For all comparisons, α = 0.05 was considered the threshold for statistical significance and adjusted with Bonferroni correction for multiple comparison. 

The segmentation performance was examined for primary tumors with the following largest dimensions: ≤2 cm (T1), 2–5 cm (T2), and ≥5 cm (T3–4).

## 3. Results

### 3.1. Segmentation Performance of Semiquantitative Parametric Maps

The Kruskal–Wallis test demonstrated significant differences among models with different input metrics (Figure 1). The χ^2^ value was 42.1 for DSC (*p* < 0.05) and 59.9 for sensitivity (*p* < 0.05); post hoc-paired Wilcoxon signed-rank test was performed and adjusted at *p* = 0.05/4 = 0.0125. For DSC, nnU-Net_BL was better than nnU-Net_PEI (*p* < 0.0125), nnU-Net_SER (*p* < 0.0125) and nnU-Net_MSI (*p* < 0.0125) but similar to nnU-Net_Comb (*p* = 0.90). For sensitivity, nnU-Net_BL was better than nnU-Net_SER (*p* < 0.0125) and nnU-Net_MSI (*p* < 0.0125) but similar to nnU-Net_PEI (*p* = 0.19) and nnU-Net_Comb (*p* = 0.58). 

### 3.2. Mask Type Comparison

A Wilcoxon signed-rank test showed similar group DSC mean ranks (*p* = 0.17) in nnUnet_Excl (without central necrosis and biopsy clips) and nnUnet_Incl (including central necrosis and biopsy clips) (Figure 2). The tumor sizes identified by the DL models were also similar to the tumor sizes of the references (Figure 2B). The segmentation of both models preserved details that corresponded accurately with their respective references (Figure 2C).

### 3.3. Segmentation Performance Using Datasets of Different Time Points

The nnU-Net_3tpt model, which was trained on a combination of data from all three time points, BL, C2, and C4, demonstrated better segmentation performance than the models trained on data from a single time point (nnU-Net_BL, nnU-Net_C2, and nnU-Net_C4) (Figure 3). The Paired Wilcoxon signed-rank test demonstrated that nnU-Net_3tpt had a better performance than nnU-Net_BL on the testing dataset of BL for both DSC (*p* < 0.05) and sensitivity (*p* < 0.05). Similarly, nnU-Net_3tpt had a better performance than nnU-Net_C2 on the testing dataset of C2 for both DSC (*p* < 0.05) and sensitivity (*p* < 0.05), and a better performance than nnU-Net_C4 on the testing dataset of C4 for both DSC (*p* < 0.05) and sensitivity (*p* < 0.05). The decrease in performance over the treatment time points in two representative subjects is shown in Figure 3D. 

The Kruskal–Wallis test demonstrated significant differences in DSC among various tumor sizes over the test set of 3tpt (χ^2^ =15.7, *p* < 0.05). The DSC had higher mean ranks for tumors larger than 2 cm but smaller than 5 cm (*p* < 0.05) in relation to tumors smaller than 2 cm. Similarly, the tumors larger than 5 cm had a higher mean rank of DSC than tumors smaller than 2 cm (*p* < 0.05), as shown in Figure 4A. In the BL test set (Figure 4B), nnU-Net_3tpt had higher mean ranks of DSC than nnU-Net_BL (*p* < 0.05) for tumors larger than 2 cm but smaller than 5 cm. For tumors that were 2 cm or smaller (*p* = 0.07) and tumors at 5 cm or larger (*p* = 0.29), the performance of the two models was similar. The higher mean ranks of DSC and sensitivity for nnU-Net_3tpt than for nnU-Net_C2 and nnU-Net_C4 across tumor sizes are presented in Appendix A.

### 3.4. Tumor Size Comparison 

The tumor size of the predicted segmentation from nnU-Net_3tpt was also evaluated using the reference standard (Figure 5). The intraclass correlation coefficient was 0.95 (*p* < 0.05) between predicted segmentation with nnU-Net_3tpt and the reference standard (Figure 5A), which demonstrated accurate tumor size estimation on the BL test set. The correlation coefficient was 0.88 at C2 test set (*p* < 0.05) and 0.73 at C4 test set (*p* < 0.05). 

## 4. Discussion

In this study, we employed an automated deep learning framework for medical imaging segmentation, nnU-Net, in conjunction with 744 datasets to develop an accurate segmentation model specifically for TNBC under treatment. We evaluated a range of semiquantitative maps from DCE-MRI as model inputs and found that the subtraction images of the initial phases from the peak arterial phases yielded the optimal contrast for model training. Additionally, using images from multiple time points during a patient’s treatment resulted in significantly better segmentation performance than using images from a single time point during treatment. This improved performance of our top-performing model likely stemmed from a greater diversity in terms of tumor size, shape, location, signal intensity patterns, and other morphological characteristics. Notably, our model accurately estimated tumor size, which is important as it is often necessary to measure changes in tumor size during treatment. 

Our results indicate that DCE subtraction images provide sufficient image information to achieve good tumor segmentation with our deep learning model. To our knowledge, most reported breast cancer segmentation models for DCE-MRI use the original multiphasic images [32,44] or the subtraction between pre-contrast and post-contrast images as the input [45]. We investigated the former approach by incorporating the entire series of DCE-MRI data at BL. However, we were able to input only about 26 patient datasets during training due to the large computational memory consumption required to store 4D image series and optimize millions of model parameters. In contrast, using subtraction images effectively reduced the dimensionality of the data from 4D to 3D and required much less memory. Even though some breast cancer studies with DCE-MRI showed the use of PEI [42,46] or MSI [47,48] parametric maps for diagnosis or treatment prediction purposes, our study showed that the use of the simple subtraction images could produce a model with better or equivalent performance for tumor segmentation compared to the use of incorporating other DCE parametric maps. Further, using subtraction images for segmentation is advantageous because they are easier to generate than the other parametric maps, whose generation may require specialized software. 

To the best of our knowledge, our study is the first to perform model training using datasets including images of patients from multiple treatment time points. The nnU-Net_3tpt model trained using this approach demonstrated better performance than the model trained using data from only a single treatment time point. By combining data from three treatment time points, the training datasets were effectively tripled. Since all the model training parameters remained identical, it is possible that the increased dataset size contributed to the improved model performance. With the expanded training dataset, our nnUNet_3tpt model exhibited performance similar to or even better than that of recent breast tumor segmentation studies. For instance, when Yue et al. evaluated model performance on a dataset of 1000 subjects (n_training = 800, n_testing = 200), their own model, Res-UNet, achieved a DSC of 0.894, and their implementation of nnU-Net achieved a DSC of 0.887 [45], whereas our nnU-Net_3tpt model achieved a DSC of 0.93 in the BL test set. Other notable studies include one in which an nnU-Net trained on a training dataset of 102 subjects achieved a DSC of 0.87 (median value, mean was not reported) on a test set of 55 subjects [49]. Additionally, a regional convolutional neural network model trained on a dataset of 241 patients, including over 10,000 slices, achieved a DSC of 0.79 on a test set of 98 patients, including approximately 9000 slices, by splitting the 3D dataset into 2D space to increase dataset size [3]. A 3D U-Net model from a full dataset of 141 subjects (n_test = 30) achieved a mean DSC of 0.78 [44]. The aforementioned models were developed for a variety of subtypes of breast cancer, not exclusively TNBC. In contrast, our nnU-Net_3tpt model, which was trained exclusively on a large sample of TNBC subjects, holds significant potential for application within the TNBC population. 

The segmentation models using masks with or without central necrosis and biopsy clips exhibited similar DSC and sensitivity, indicating that our models based on the nnU-Net framework are flexible and stable. To the best of our knowledge, most published studies on breast tumor segmentation employed reference masks that included central necrosis and biopsy clips [3,28,44,50]. However, necrosis and biopsy clips may need to be excluded for certain applications, such as functional tumor volume measurements. Our findings indicate that our models can directly output both types of masks without added processing, and segmentation performance is similar with and without exclusion of central necrosis and biopsy clips. In contrast to a recent study [45], in which the tumor was segmented first and then an intensity-based method was applied to delineate a low signal intensity of necrosis within the segmented tumor region, our fully automated model provides an easier approach.

Our nnU-Net_3tpt model had good performance but can be further improved in several aspects. First, nnU-Net_3tpt produced increasingly better segmentation results for tumors with larger sizes, which may explain the better performance at BL than at C2 and C4 because tumors at BL are untreated and tend to be larger. A similar trend was noticed in other studies [45,49,51]. The performance of our model on smaller tumors could be improved by including a more diverse range of samples, and the generalizability of the model could be validated by including public datasets for more comprehensive training and independent testing. Second, the nnU-Net_3tpt model failed to identify tumors smaller than 2 cm in several instances. To avoid such failures, it may be necessary to refine the nnU-Net framework configuration by modifying the loss function to prioritize false negatives. In our training, the model training loss term is guided by DSC, which emphasizes both sensitivity and precision. Other metrics for training loss may be designed to penalize false negatives with heavier weighting. DSC may not be optimal to address the signal heterogeneity of TNBC. An alternative metric for training loss is focal Dice loss, which could alleviate the imbalance between empirically defined subtypes [52]. Models that extract semantic features could integrate spatial information to improve sensitivity to tumors at smaller sizes and tissue boundaries, making it worthwhile to validate their efficacy in the TNBC population [53,54]. Finally, a systematic comparison of our model to the conventional models using the same datasets would better evaluate our model. In addition to the static imaging features used in our study, integrating tumor-specific dynamic information into the nnU-Net framework could also help to reduce false positives [50].

## 5. Conclusions

We developed a fully automated, high-performance segmentation model for TNBC patients using deep learning and a large cohort of DCE-MRI images acquired longitudinally over the patients’ treatment course. Of the various types of images used for model training, we found that the simple subtraction images had the best performance. Our model was also capable of reliably segmenting tumors with either exclusion or inclusion of the central necrosis and biopsy clips. The performance of our model, especially for small tumors, may be further improved in a future investigation. 

## Figures and Tables

**Figure 1 cancers-15-04829-f001:**
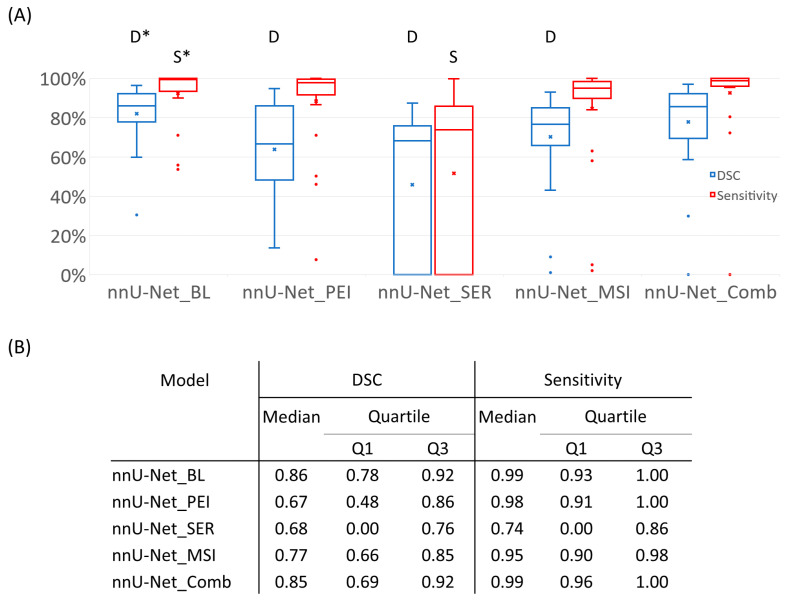
Segmentation performance of nnU-Net models with different combinations of BL DCE images and semiquantitative parametric maps. The DSC and sensitivity were measured at the subject level using manually labeled masks as the reference standard and were then averaged across the BL test set. (**A**) The boxplots of each set of results, first and third quartiles (lower and upper ends of box, respectively), the min and max limits (whiskers) at 1.5 interquartile away from the first and third quartiles; median (horizontal line in box), mean (x), and outliers (discrete data points) were presented. (The letters above the boxplots indicated statistical significance between that metric and the reference metric, which was labeled with the same letter and an asterisk on top). (**B**) The detailed quantitative results used for the boxplots in (**A**).

**Figure 2 cancers-15-04829-f002:**
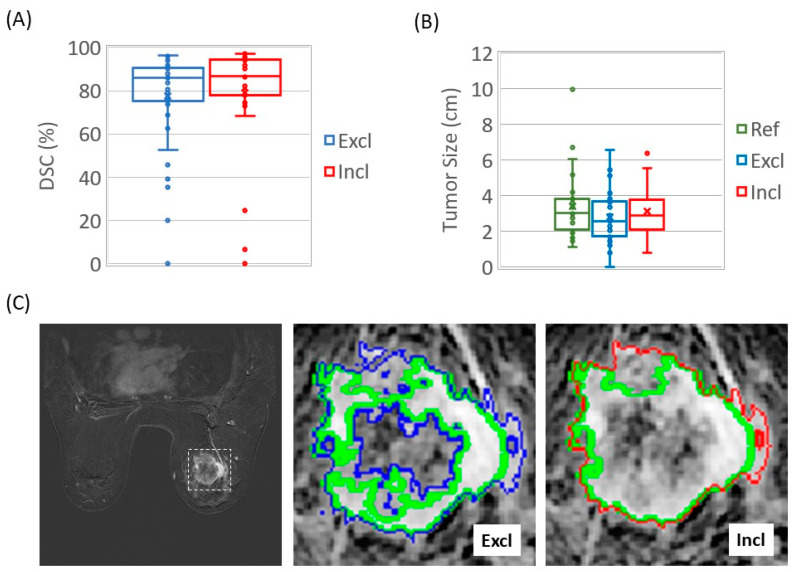
Automated tumor segmentation with and without inclusion of central necrosis and biopsy clips. (**A**) nnU-Net_Excl and nnU-Net_Incl on the same test cases had similar DSCs (*p* = 0.27). (**B**) Tumor sizes based on reference masks (ref: green) were similar to those estimated with nnU-Net_Excl (*p* = 0.14) and nnU-Net_Incl (*p* = 0.58). (**C**) Automated masks without inclusion (Excl: blue) and with inclusion (Incl: red) of central necrosis and biopsy clips overlaying corresponding reference (green) mask of a representative subject. The subimage within the dashed box has been zoomed in and displayed as the background in the Excl and Incl images.

**Figure 3 cancers-15-04829-f003:**
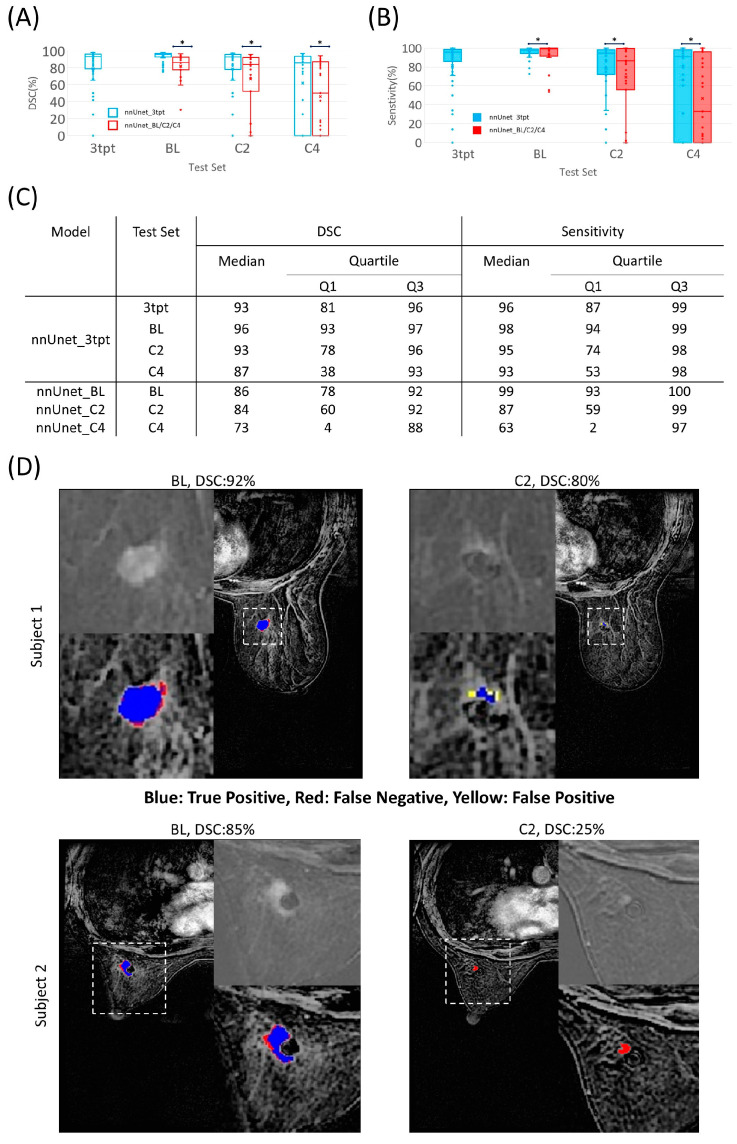
Segmentation performance of nnU-Net models using data from various time points. DSC (**A**) and sensitivity (**B**) of the different models applied to the corresponding testing dataset. Blue bars on top of the dataset indicate significant difference in paired Wilcoxon signed-rank test (*p* < 0.05, indicated by black asterisks). (**C**) The detailed quantitative results used for the boxplots (**A**,**B**). (**D**) Two representative subjects and the predicted segmentation performance using the nnU-Net_3tpt model. The reference mask is the union of blue and red masks.

**Figure 4 cancers-15-04829-f004:**
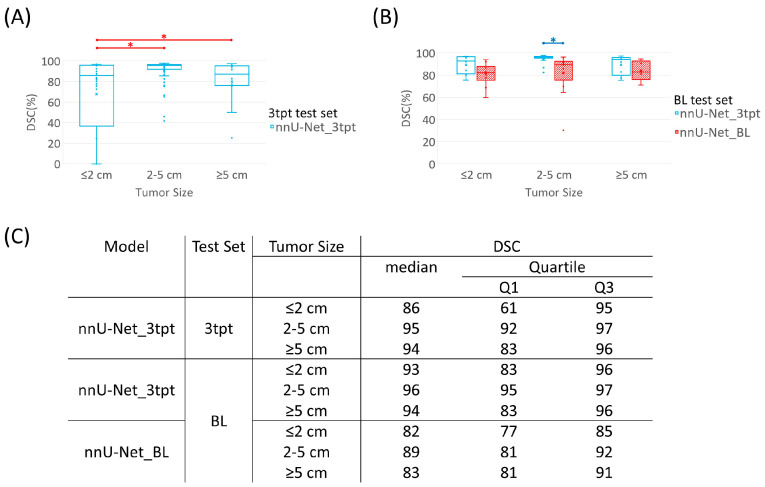
Segmentation performance of nnU-Net models by tumor size. (**A**) DSC of nnU-Net_3tpt across various tumor sizes in the 3tpt test set. Red bars indicate significant difference in unpaired Wilcoxon rank-sum test adjusted at *p* < 0.016 (indicated by red asterisks). (**B**) DSC of nnU-Net_3tpt and nnU-Net_BL applied on BL test sets across various tumor sizes. Blue bar indicates statistically significant difference in paired Wilcoxon signed-rank test (*p* < 0.05, indicated by blue asterisks). (**C**) The detailed quantitative results used for the boxplots in (**A**,**B**).

**Figure 5 cancers-15-04829-f005:**
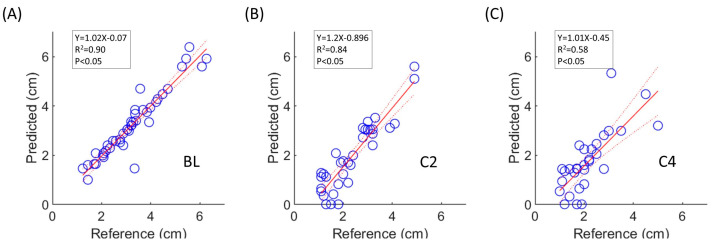
Comparison of tumor size between predicted segmentation using nnU-Net_3tpt and reference tumor mask. Shown are linear relationships between tumor size of predicted segmentation and reference at BL (**A**), C2 (**B**), and C4 (**C**). The best-fit linear regression was represented by the solid line, while the 95% confidence interval bands were denoted by dashed lines.

**Table 1 cancers-15-04829-t001:** Patients’ characteristics of all datasets included in this study and datasets from baseline (BL), after 2 cycles (C2), and after 4 cycles (C4) of neoadjuvant systemic therapy.

Characteristic	All Datasets	BL	C2	C4
No. of datasets	744	285	207	252
Age, mean ± SD, years	50 ± 11	50 ± 11	50 ± 11	50 ± 11
Longest tumor diameter, mean ± SD, cm	2.7 ± 1.6	3.4 ± 1.5	2.6 ± 1.4	2.1 ± 1.5
Clinical stage, *n* (%)				
I	96 (13)	37 (13)	29 (14)	30 (12)
II	542 (73)	210 (74)	148 (72)	184 (73)
III	106 (14)	38 (13)	30 (14)	38 (15)
T category, *n* (%)				
T1	139 (19)	54 (19)	39 (19)	46 (18)
T2	509 (68)	195 (68)	141 (68)	173 (69)
T3	83 (11)	31 (11)	23 (11)	29 (12)
T4	13 (2)	5 (2)	4 (2)	4 (2)
N category, *n* (%)				
N0	490 (66)	188 (66)	139 (67)	163 (65)
N1	171 (23)	67 (24)	44 (21)	60 (24)
N2	26 (3)	9 (3)	8 (4)	9 (4)
N3	57 (8)	21 (7)	16 (8)	20 (8)

**Table 2 cancers-15-04829-t002:** Constructed models and their inputs. Sub, subtraction image.

Model Name	Input Dataset	DCE Metrics
nnU-Net_BL	BL	Sub
nnU-Net_PEI	BL	PEI
nnU-Net_SER	BL	SER
nnU-Net_MSI	BL	MSI
nnU-Net_Comb	BL	Sub + PEI + MSI + SER
nnU-Net_C2	C2	Sub
nnU-Net_C4	C4	Sub
nnU-Net_3tpt	BL + C2 + C4	Sub
nnU-Net_Excl	BL	Sub
nnU-Net_Incl	BL	Sub

## Data Availability

Data are available from the corresponding author upon request.

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
