# Peer review of "Deep Learning for Fully Automatic Tumor Segmentation on Serially Acquired Dynamic Contrast-Enhanced MRI Images of Triple-Negative Breast Cancer"

_cancers, 2023, doi:10.3390/cancers15194829_

Round 1

Reviewer 1 Report

In this paper, authors developed a fully automated and high-performance segmentation model of triple-negative breast cancer using a self-configurable deep learning framework and a large set of dynamic contrast-enhanced MRI images acquired serially over the patient treatment course. The authors did good work and were interested for the readers. The following review comments are recommended, and authors are invited to explain and modify.

1 Why is it important using the subtraction between pre-contrast and post-contrast MRI images? Is it not obvious abnormal pixels in post-contrast?  

2 “A total of 301 patients with biopsy-confirmed stage I-III TNBC were included”, why did not authors use public available data of TCIA Images?

3 What is logic to be used nnU-Net base model?

4  An introduction is an important road map for the rest of the paper that should be consist of an opening hook to catch the researcher's attention, relevant background study, and a concrete statement that presents main argument but your introduction lacks these fundamentals, especially relevant background studies that needs to be improved. This related work is just listed out without comparing the relationship between this paper's model and them; only the method flow is introduced at the end; and the principle of the method is not explained. To make soundness of your study must include these latest works:

I (2023). Stimulated Raman Scattering Microscopy Enables Gleason Scoring of Prostate Core Needle Biopsy by a Convolutional Neural Network. Cancer Research, 83(4), 641-651. doi: 10.1158/0008-5472.CAN-22-2146

II (2021). Atractylenolide I enhances responsiveness to immune checkpoint blockade therapy by activating tumor antigen presentation. The Journal of Clinical Investigation, 131(10). doi: 10.1172/JCI146832

III (2022). The Effect of Sevoflurane on the Proliferation, Epithelial-Mesenchymal Transition (EMT) and Apoptosis in Human Breast Cancer Cells. Journal of Biological Regulators and Homeostatic Agents, 36(3), 583-592. doi: 10.23812/j.biol.regul.homeost.agents.20223603.66

IV (2022). Automatic interpretation and clinical evaluation for fundus fluorescein angiography images of diabetic retinopathy patients by deep learning. British Journal of Ophthalmology, 2022-321472. doi: 10.1136/bjo-2022-321472

V (2023). Identifying Malignant Breast Ultrasound Images Using ViT-Patch. Applied Sciences, 13(6), 3489. doi: 10.3390/app13063489"

5 “During the model training, the input data were first preprocessed”, authors need give all details of image preprocessing step.

6 “Manually segmented the tumors in consensus using an in-house MATLAB-based software”, should be shown sample raw images and corresponding labels.

7 There should be some Tables in “Results” section.

8 When writing phrases like “The DSC and sensitivity were calculated and averaged across all subjects as the metrics for overall performance”, it must cite related work in order to sustain the statement 

9 How to optimize hyperparameters during model training?

10 Authors should mention the implementation challenges.

11 Authors should discuss the limitations and future works of the developed system elaborately.

12 Results are not clearly compared with the state-of-art. This point is fundamental in scientific research.

13 Moreover, it should be noticed that the clinical appliance has to be decided by medicals since the existing differences between the real image and the one generated by the proposed model could be substantial in the medical field.

Minor editing of English language required.

Author Response

R1.1 Why is it important using the subtraction between pre-contrast and post-contrast MRI images? Is it not obvious abnormal pixels in post-contrast?  

Evaluating the subtraction images derived from pre-contrast and post-contrast DCE is standard clinical practice. Prior research has shown that these subtraction images are effective in delineating lesion boundaries and extent of the lesion, particularly in breast cancer imaging analysis (doi: 10.1259/bjr/98435332; 10.1371/journal.pone.0234800) and segmentation studies (doi: 10.1002/nbm.3132).  Below, we display both the subtraction and post-contrast images to highlight the enhanced tumor-tissue boundary definition and reduced background tissue intensity in the subtraction image.

Figure R1. the subtraction images between pre-contrast and post-contrast DCE and the post-contrast images of the identical subjects that are shown in new Figure 2. Images from the same subject were adjusted with the same window level and depth. 

R1.2 “A total of 301 patients with biopsy-confirmed stage I-III TNBC were included”, why did not authors use public available data of TCIA Images?

The objective of this study was to develop an automated breast tumor segmentation system specifically for the TNBC population. To the best of our knowledge, the internal cohort we examined in this study is the largest of its kind. We ensured a high level of uniformity in the dataset to minimize data heterogeneity, which might arise from various imaging sites, field strengths, spatial/temporal resolutions, and other factors. However, we agree with the reviewer on the importance of incorporating public data to enhance the generalizability of our model, which was discussed in the revised Discussion section.

R1.3 What is logic to be use nnU-Net base model?

Since the introduction of nnU-Net in 2021, it has been widely recognized and employed in automated segmentation studies across various imaging modalities and disease models. We chose to build our model using this framework largely because of its established flexibility and accuracy.

R1.4  An introduction is an important road map for the rest of the paper that should be consist of an opening hook to catch the researcher's attention, relevant background study, and a concrete statement that presents main argument but your introduction lacks these fundamentals, especially relevant background studies that needs to be improved. This related work is just listed out without comparing the relationship between this paper's model and them; only the method flow is introduced at the end; and the principle of the method is not explained. To make soundness of your study must include these latest works:

I (2023). Stimulated Raman Scattering Microscopy Enables Gleason Scoring of Prostate Core Needle Biopsy by a Convolutional Neural Network. Cancer Research, 83(4), 641-651. doi: 10.1158/0008-5472.CAN-22-2146

II (2021). Atractylenolide I enhances responsiveness to immune checkpoint blockade therapy by activating tumor antigen presentation. The Journal of Clinical Investigation, 131(10). doi: 10.1172/JCI146832

III (2022). The Effect of Sevoflurane on the Proliferation, Epithelial-Mesenchymal Transition (EMT) and Apoptosis in Human Breast Cancer Cells. Journal of Biological Regulators and Homeostatic Agents, 36(3), 583-592. doi: 10.23812/j.biol.regul.homeost.agents.20223603.66

IV (2022). Automatic interpretation and clinical evaluation for fundus fluorescein angiography images of diabetic retinopathy patients by deep learning. British Journal of Ophthalmology, 2022-321472. doi: 10.1136/bjo-2022-321472

V (2023). Identifying Malignant Breast Ultrasound Images Using ViT-Patch. Applied Sciences, 13(6), 3489. doi: 10.3390/app13063489"

Our introduction provides some basic background on the nature of TNBC breast cancer and the importance/challenges of its image segmentation. We also introduced the model-based and deep-learning-based methods, noting in particular the challenges of building a model with the DCE dataset and our proposed method.

We value the reviewer's suggestions on the references. However, we carefully examined the references cited by the reviewer and feel that they are either cellular works or deep learning applications other than segmentation, which are lack of enough relevance to our work. In fact, Reference III appears to be invalid, with an erroneous title and DOI. Instead of the suggestions, we have incorporated new references, specifically #30, #31, and #32, to enhance the background knowledge on breast tumor segmentation. The methodology of using nnU-Net was previously addressed in response R1.3.

R1.5 “During the model training, the input data were first preprocessed”, authors need give all details of image preprocessing step.

We clarified the question and provided details of the preprocessing in the revision.

R1.6 “Manually segmented the tumors in consensus using an in-house MATLAB-based software”, should be shown sample raw images and corresponding labels.

We have provided examples in Figures 2 and 4 to help clarify the reviewer’s question.

R1.7 There should be some Tables in “Results” section.

Thank you for the suggestion. We have listed all the quantitative results used for the boxplots in Figures 1-3.

R1.8 When writing phrases like “The DSC and sensitivity were calculated and averaged across all subjects as the metrics for overall performance”, it must cite related work in order to sustain the statement. 

This sentence outlines the metrics we utilized to assess model performance. We chose DSC and sensitivity because they are broadly recognized in the field of image segmentation. The definition of these metrics is now cited in the updated reference # 43.

R1.9 How to optimize hyperparameters during model training?

In our study, we used the hyperparameters predefined and auto-configured by the nnU-Net framework without any manual adjustments. These hyperparameters can be categorized into "rule-based" parameters—automatically set based on the training dataset—and "fixed" pre-trained parameters, which are empirically determined by nnU-Net. As highlighted in the original nnU-Net publication, models trained with these automatically optimized hyperparameters outperformed most existing methods across 23 public datasets in segmentation competitions. Using these auto-tuned hyperparameter sets, we achieved high DSC and sensitivity, as depicted in the updated Figure 2. Furthermore, we analyzed the training and validation losses for all models (see Figure R2) and observed consistent decreasing trends across epochs, indicating that our models neither overfitted nor underfitted the data. Given these metrics, we believe that the chosen hyperparameter sets are suitable for our segmentation objectives.

Figure R2: The training and validation losses of randomly chosen one of the five folds in the construction of nnU-Net_3tpt over 1000 epochs, the loss term is the summation of cross-entropy and Dice loss.

Figure R2: The training and validation losses of randomly chosen one of the five folds in the construction of nnU-Net_3tpt over 1000 epochs, the loss term is the summation of cross-entropy and Dice loss.

R1.10 Authors should mention the implementation challenges.

The technical challenges can be categorized into two main areas:

  1. The successful application of existing deep learning segmentation models, which are trained on one dataset but applied to other datasets is often very limited. This is typically due to hyperparameter tuning being dataset-specific, which means adapting the model can complicate clinical workflows. To address this, we propose using the nnU-net method. This U-Net based segmentation approach automatically optimized hyperparameters according to “data fingerprint” (as detailed in response to R3.8) and according to empirical configurations pre-trained across diverse applications.
  2. The availability of DCE data for the TNBC population, especially from multiple time points throughout the patient treatment, is limited. The challenge lies in determining how to integrate this data and devise a model applicable across the different time points of the patient treatment.

We've underscored these challenges in the revised Introduction.

R1.11 Authors should discuss the limitations and future works of the developed system elaborately.

We have expanded the discussion of limitations and future works with more details and references.

R1.12 Results are not clearly compared with the state-of-art. This point is fundamental in scientific research.

Thank you for the comment. As we addressed Discussion, we compared our model to several works in the field of breast tumor segmentation and our results were comparable or superior. While a direct, "apple-to-apple" comparison with other models may not be valid due to differences in datasets and patient populations, the robust performance of our model underscores the novelty of our approach.

R1.13 Moreover, it should be noticed that the clinical appliance has to be decided by medicals since the existing differences between the real image and the one generated by the proposed model could be substantial in the medical field.

We agreed with the comments regarding the quality of translating the proposed model for clinical applications. In this study, the proposed model was trained with reference segmentations by experienced radiologists, and the model performance was quantitatively evaluated with standard metrics. Future studies may include comparison of the mode performance with that by the practicing radiologists or different state-of-the-art models.

Reviewer 2 Report

This manuscript describes the development of a fully automated and high-performance segmentation model of triple-negative breast cancer using a self-configurable deep-learning framework and a large set of dynamic contrast-enhanced MRI images of patients from multiple treatment time points. The authors reported that the nnU-324 Net_3tpt model trained using this approach demonstrated better performance than the model trained using data from only a single treatment time point. Moreover, the authors also found that the simple subtraction images had the best performance among the various types of images used for model training. It is a very well-written paper and suitable for publication in Cancers. One minor thing related to the formatting of references #1 and 45 should be addressed (page numbers/manuscript numbers should be added).

Author Response

R2.1.This manuscript describes the development of a fully automated and high-performance segmentation model of triple-negative breast cancer using a self-configurable deep-learning framework and a large set of dynamic contrast-enhanced MRI images of patients from multiple treatment time points. The authors reported that the nnU-324 Net_3tpt model trained using this approach demonstrated better performance than the model trained using data from only a single treatment time point. Moreover, the authors also found that the simple subtraction images had the best performance among the various types of images used for model training. It is a very well-written paper and suitable for publication in Cancers. One minor thing related to the formatting of references #1 and 45 should be addressed (page numbers/manuscript numbers should be added).

Thank you for the suggestion regarding the details to enhance our work. We have reformatted reference #1 in line with the journal's requirements. However, the page or article ID for the original reference #45 (now #50) is not available on the Journal of Magnetic Resonance Imaging website. For the time being, we have retained it as is, but we've ensured and verified its validity.

Reviewer 3 Report

This paper presents a deep-learning-based breast cancer segmentation method in contrast-enhanced MRI images. The method is sound. The experiments show the effectiveness of the proposed method. However, there are some concerns about this study. This paper needs to be refined before it can be accepted.

1. The technical challenge of the proposed method is not very unclear, especially for the deep learning technologies that this study aims at.

2. It is better to summarize the main contributions in this study at the end of the introduction section.

3. It is to introduce the inclusion and exclusion criterion in data collection.

4. More studies on medical image segmentation need to be cited, e.g.: Vessel Contour Detection in Intracoronary Images via Bilateral Cross-Domain Adaptation; Densely connected deep convolutional encoder-decoder network for nasopharyngeal carcinoma segmentation; Progressive Perception Learning for Main Coronary Segmentation in X-ray Angiography.

5. Why does the proposed method focus on nnU-net ?

6. It is better to briefly introduce the definition of true positive, false negative, false positive, and true negative.

7. Why is alpha set 0.05 in Line 215, but the threshold of p-values are different in Section 3.1. 

8. The meaning of "data fingerprint" is not clear.

9. It is not very easy to understand Fig.2. Please refine this figure.

10. Some grammatical errors.

moderate refinement

Author Response

This paper presents a deep-learning-based breast cancer segmentation method in contrast-enhanced MRI images. The method is sound. The experiments show the effectiveness of the proposed method. However, there are some concerns about this study. This paper needs to be refined before it can be accepted.

R3.1. The technical challenge of the proposed method is not very unclear, especially for the deep learning technologies that this study aims at.

The technical challenges can be categorized into two main areas:

  1. The successful application of existing deep learning segmentation models, which are trained on one dataset but applied to other datasets is often very limited. This is typically due to hyperparameter tuning being dataset-specific, which means adapting the model can complicate clinical workflows. To address this, we propose using the nnU-net method. This U-Net based segmentation approach automatically optimized hyperparameters according to “data fingerprint” (as detailed in response to R3.8) and according to empirical configurations pre-trained across diverse applications.
  2. The availability of DCE data for the TNBC population, especially from multiple time points throughout the patient treatment, is limited. The challenge lies in determining how to integrate this data and devise a model applicable across the different time points of the patient treatment.

We've underscored these challenges in the revised Introduction.

R3.2. It is better to summarize the main contributions in this study at the end of the introduction section.

We have revised the last paragraph of introduction accordingly.

R3.3. It is to introduce the inclusion and exclusion criterion in data collection.

We have provided a reference to the inclusion/exclusion criteria that were used in our study. The same criteria are provided below:

TNBC was defined from standard pathologic assays as negative for ER and PR (<10% tumor staining) and negative for HER2 (immunohistochemistry (IHC) score < 3, gene copy number not amplified). Patients with stage IV disease prior to the initiation of chemotherapy, or who have had a prior excisional biopsy of the primary invasive breast cancer, or who are not eligible for taxane and/or anthracycline-based chemotherapy regimens, were excluded from this study.”

R3.4. More studies on medical image segmentation need to be cited, e.g.: Vessel Contour Detection in Intracoronary Images via Bilateral Cross-Domain Adaptation; Densely connected deep convolutional encoder-decoder network for nasopharyngeal carcinoma segmentation; Progressive Perception Learning for Main Coronary Segmentation in X-ray Angiography.

We appreciated the reviewer’s suggestions. We included the latter two suggested works as new references #55 and #56 in the discussion. Even though they were not on breast tumor segmentation, the models using semantic features would be useful to improve accuracy to tumors at smaller size and preserving boundary details, which provides potential solutions to our limitations.

R3.5. Why does the proposed method focus on nnU-net ?

Please see our response to R3.1

R3.6. It is better to briefly introduce the definition of true positive, false negative, false positive, and true negative.

We have described the details of those definitions in the method section 2.5.

R3.7. Why is alpha set 0.05 in Line 215, but the threshold of p-values are different in Section 3.1. 

In the original study, the alpha value was set at 0.05 for all statistical tests, but a Bonferroni correction was applied for multiple comparisons, resulting in a reduced p-value. However, as addressed in response R4.1, the original statistical approach did not account for the non-normal distribution of the results, which disqualified the ANOVA or t-test. We have since re-evaluated the results using the Wilcoxon rank test and the Kruskal-Wallis test. The associated p-values were set at 0.05 and adjusted using the Bonferroni criterion.

R3.8. The meaning of "data fingerprint" is not clear.

The concepts of “data fingerprint” was defined by the original nnU-net publication [reference #36]. As we have briefly explained in section “2.4 Automatic Segmentation Framework”, its context includes image size, voxel size, median shapes, signal intensity distribution, spacing distribution, number of classes, number of training cases and modality.

R3.9. It is not very easy to understand Fig.2. Please refine this figure.

Thank you for the comment. We have revised Figure 2 and its caption accordingly.

R3.10. Some grammatical errors.

Thanks for help us on the details, we have revised the entire manuscript carefully on its format and grammars.

Reviewer 4 Report

Dear Authors, congrats for the very interesting paper. I have a single serious perplexity.

-at line 103 you say that the models were evaluated in terms of DSC and sensitivity.

At my knowledge the value of a DSC ranges from 0 (no overlap) to 1 (complete overlap)

Same thing applies to sensitivity, the True Positive Rate can be 1 at max.

So... how can you write at line 226 that for nnU-Net_BL you have a DSC of 92%+-14% (then between 78% and 106%)?

The same observation applies to the segmentation performances in fig2. At line 248 you write that the average sensitivity was 82+-30%, but a sensitivity oh 112% is out-of-range by definition.

So please clarify this very important point.

Author Response

Thank you for bringing this important point in our original work. We've restructured the results section, introducing new statistical analyses and figures. The original results were skewed, with zeros evident in Figure R3. We've observed that these zero sensitivity results correlate with tumor sizes smaller than 2 cm. We've since performed comparisons using the Wilcoxon rank test and the Kruskal-Wallis test. We've detailed this new statistical approach in the revised section 2.5, and have made corresponding changes in the results.

Figure R3, the histogram of nnU-Net_3tpt sensitivity result on the test set of BL and C4, the same results were also used for boxplot in Figure 2B in the revised manuscript.
Figure R3, the histogram of nnU-Net_3tpt sensitivity result on the test set of BL and C4, the same results were also used for boxplot in Figure 2B in the revised manuscript.

Round 2

Reviewer 1 Report

Accept

Reviewer 3 Report

No further question.

minor refinement

Reviewer 4 Report

Thanks for the clarification.